# Hyperthermia in Combination with Emerging Targeted and Immunotherapies as a New Approach in Cancer Treatment

**DOI:** 10.3390/cancers16030505

**Published:** 2024-01-24

**Authors:** Tine Logghe, Eke van Zwol, Benoît Immordino, Kris Van den Cruys, Marc Peeters, Elisa Giovannetti, Johannes Bogers

**Affiliations:** 1Elmedix NV, Dellingstraat 34/1, 2800 Mechelen, Belgium; 2Cancer Pharmacology Lab, Fondazione Pisana per la Scienza, San Giuliano, 56017 Pisa, Italy; 3Institute of Life Sciences, Sant’Anna School of Advanced Studies, 56127 Pisa, Italy; 4Department of Oncology, Antwerp University Hospital, 2650 Edegem, Belgium; 5Department of Medical Oncology, Amsterdam UMC, Location Vrije Universiteit, Cancer Center Amsterdam, 1081 HV Amsterdam, The Netherlands; 6Laboratory of Cell Biology and Histology, Faculty of Medicine and Health Sciences, University of Antwerp, 2610 Antwerp, Belgium

**Keywords:** hyperthermia, cancer, immunotherapy, emerging therapies, targeted therapies, PD-1/L1, BiTEs, VEGF

## Abstract

**Simple Summary:**

This manuscript discusses the ongoing challenge of cancer as a leading global cause of death despite advancements in therapies. It highlights the role of hyperthermia (HT) as a modality in cancer treatment, particularly its effectiveness as a sensitizer and its impact on cancer–immunity processes and oncogenic pathways. The article notes the recent focus on immunotherapy (IT) and targeted therapy (TT) in cancer research, both in academia and pharmaceutical companies. The main focus of the manuscript is to explore potential therapies that can enhance the effects of HT by targeting molecular pathways. The ultimate goal is to pave the way for future research and clinical trials, aiming to harness the synergistic potential of combining emerging IT and TT with HT for improved outcomes.

**Abstract:**

Despite significant advancements in the development of novel therapies, cancer continues to stand as a prominent global cause of death. In many cases, the cornerstone of standard-of-care therapy consists of chemotherapy (CT), radiotherapy (RT), or a combination of both. Notably, hyperthermia (HT), which has been in clinical use in the last four decades, has proven to enhance the effectiveness of CT and RT, owing to its recognized potency as a sensitizer. Furthermore, HT exerts effects on all steps of the cancer–immunity cycle and exerts a significant impact on key oncogenic pathways. Most recently, there has been a noticeable expansion of cancer research related to treatment options involving immunotherapy (IT) and targeted therapy (TT), a trend also visible in the research and development pipelines of pharmaceutical companies. However, the potential results arising from the combination of these innovative therapeutic approaches with HT remain largely unexplored. Therefore, this review aims to explore the oncology pipelines of major pharmaceutical companies, with the primary objective of identifying the principal targets of forthcoming therapies that have the potential to be advantageous for patients by specifically targeting molecular pathways involved in HT. The ultimate goal of this review is to pave the way for future research initiatives and clinical trials that harness the synergy between emerging IT and TT medications when used in conjunction with HT.

## 1. Introduction

### 1.1. Immunotherapy and Targeted Therapy in Cancer Research

Over the past decade, the field of cancer treatment has undergone a swift and significant transformation. Traditional therapies, such as chemotherapy (CT) and radiotherapy (RT), tailored to specific tumor types and patient staging have been optimized in recent years, leading to improved survival rates [1]. Despite these improvements in the effectiveness of conventional therapies, the median survival time remains very low for several types of cancer [2]. Relatively new treatments like immunotherapy (IT) and targeted therapy (TT) are currently emerging and are increasingly applied in the clinic. This shift is also noticeable in the pipeline of pharmaceutical companies, where the number of investigated CTs is negligible compared to the expansion of tested IT and TT.

All types of compounds capable of engaging the immune system and weaponizing it against tumor cells are classified under IT. Within this broad definition, four categories are identified: (1) cancer vaccines, (2) immune checkpoint inhibitors (ICIs), (3) bispecific T-cell engagers (BiTEs), and (4) adoptive cell therapies.

All of these have a mechanism of action (MoA) in common, namely the elicitation of a T-cell-based immune response against the tumor. The difference lies in how T-cell activation is incited [1,3,4]. A schematic representation of all types of ITs is depicted in Figure 1.

TT is a treatment that hampers carcinogenesis through interference with certain molecules. Due to its directed “target” towards cancer cells, it is expected that TT is both more effective and less harmful than systemic CT. Currently, two main categories of TT exist: (1) small molecule inhibitors and (2) monoclonal antibodies (mAbs) [4,5,6].

### 1.2. Hyperthermia as a Potential Adjuvant

Hyperthermia (HT), also known as thermal therapy or thermotherapy, is a treatment modality that has the potential to have a synergistic effect in combination with other treatments. HT has various effects on the body and has long been deemed a RT and CT sensitizer in several types of cancer due to its pleiotropic mechanisms [7]. This approach takes advantage of the sensitivity of cancer cells to heat, while attempting to minimize damage to healthy surrounding tissues. Broad classifications of HT can be made based on the area treated or the temperature used. (1) Local HT usually uses external energy sources like microwave, radiofrequency, ultrasound, or infrared devices to heat the tumor region. (2) Regional HT is applied to larger areas of the body, such as a whole limb or an organ, and can be achieved using special heating devices or by perfusing heated fluids in the area. (3) Whole-body HT, wherein the entire body is heated, can be used for treatment of metastatic cancer [8]. Temperatures used in HT can vary, and classification by temperature is not straightforward. Temperatures in the lower range (fever-range) are generally considered immune-boosting, while higher temperatures around 43 °C aim towards direct cell killing or ablation. A schematic overview of sensitizing effects initiated by HT are depicted in Figure 2 [9,10,11].

Further research is needed to thoroughly explore the interactions between cutting-edge ITs or TTs and HT, highlighting an area with great potential for future investigation. As one crucial pathway of mild HT involves the stimulation of the immune system and response, several of the implicated pathways can be of importance for potential synergy. These rely on the general idea of making the tumor “hot” or immune active. When a tumor is more visible to the immune system through the infiltration of tumor-infiltrating lymphocytes (TILs) and other immune cells, it will make the consequent response faster and more effective [12].

### 1.3. Effects of Hyperthermia on the Immune System and Responses

Various mechanisms underlie the impact of HT on the immune system, all of which contribute to the creation of a tumor environment characterized by heightened immune activity.

#### 1.3.1. Hyperthermia Induces the DNA Damage Response and Inhibits Repair Mechanisms

Well-described effects of HT include the induction of the DNA damage response and interference with several DNA repair pathways. HT may induce damage via single- and double-stranded breaks (SSBs and DSBs, respectively), especially when combined with other DNA damage-based treatment modalities such as RT [13,14].

Such DSBs could occur due to HT (<41.5 °C) hindering the repair of faulty bases through the phosphorylation of histone H2AX, resulting in phospho-H2AX (γH2AX) foci [15]. This kind of foci resembles RT-induced foci (IRIF), which are normally formed as a response to DSBs. Moreover, the number of γH2AX foci has been deemed proportional to the thermal dose and cell killing effect [14,15,16].

A key factor in the indirect effect of HT triggering a DNA damage response is oxidative stress. The induction of oxidative stress due to HT is caused by a perturbance of the balance between production of reactive oxygen species (ROS) and their metabolization via antioxidants. Under heat stress, increases in ROS and other free radicals have been observed in cells, leading to damage to cellular macromolecules, including DNA. Furthermore, oxidative stress can affect the integrity of mitochondria, which are an important contributor to the redox status of a cell, thereby leading to DNA damage.

A last major mechanism indirectly triggered by HT-induced signaling is cell cycle checkpoint activation. During the cell cycle, there are three important checkpoints, being the G1, G2/M and M phases. Two master kinases, ATM and ATR, are responsible for sensing damage and the consequent response [14]. ATR is both triggered as a response to SSBs as well as in reaction to ATM activation due to DSBs. When activated, these kinases will mobilize repair factors, resulting in the stimulation of a specific DNA repair pathway [13,17]. Upon the administration of HT, it has been reported that ATR and ATM are activated and can phosphorylate, among others, Chk1 and Chk2, respectively. Via further downstream signaling, they cause the cell cycle to be halted at the G1/S and G2/M phases [14,18]. Reportedly, HT also inflicts abrogation on the M checkpoint, but the underlying mechanisms have not yet been identified [14].

Several repair pathway mechanisms exist, each specific for a certain type of damage. HT has been reported to be capable of interfering with various DNA repair pathways, which is also one of the reasons why this treatment is considered a potent CT sensitizer. Inhibition of repair pathways ensures that damage inflicted by CT agents is maintained [13,14].

In summary, any kind of tumoral DNA damage will begin to accumulate under HT conditions due to malfunctioning repair pathways and the possible induction of SSBs and DSBs. This causes the formation of tumor neoantigens that are specific for cancer cells and are absent in healthy somatic cells [19,20,21]. Studies have been performed showing that a higher tumor mutation burden, a proxy marker of neoantigen formation, has a significant effect on the response rate to programmed death protein 1 and programmed death ligand 1 (PD-1/PD-L1) blockade [12,21]. Therefore, HT could significantly enhance the efficacy of PD-1/PD-L1 therapy through the indirect promotion of neoantigen formation.

#### 1.3.2. Hyperthermia Induces Immunogenic Cell Death through HSP Release, ER Stress, and ROS

For a very long time, it was thought that most regulated cell death mechanisms, such as apoptosis, did not influence immunogenicity. However, over the past decade, this notion has been refuted, as numerous studies have shown that danger signaling occurs not only in necrotic and other non-apoptotic cancer cells but also in apoptotic ones. This type of cell death is known as immunogenic cell death (ICD) and is characterized by specific features that increase the immunogenicity of dying cells [22,23]. The simultaneous induction of ROS and endoplasmic reticulum (ER) stress are the most important factors required to trigger ICD.

There are two ways in which ICD is achieved (Figure 3). The first (type I ICD) is a result of collateral stress effects on the ER and is constituted by effectors that mainly target cytosolic proteins, membrane channels, or other cellular components that can indirectly impact the ER. The second (type II ICD) is established by mechanisms primarily targeting the ER itself and thereby causing stress [22,23,24]. Consequently, in both types of ICD, the expression of damage-associated molecular patterns (DAMPs) will follow, which will elicit an adaptive immune response. A few essential DAMPs, such as HMGB1-release, surface-exposed CRT, HSP70/HSP90, and secreted ATP, have been identified [22,25,26]. Additionally, ICD could also lead to the exposure of certain cancer- or tumor-associated antigens (CAAs & TAAs), which will be recognized by immune cells and contribute to a response that is boosted by DAMP danger signaling [12,24,27].

HT can also be considered as a type of ICD inducer due to various reasons. The first one is that HT induces ER stress. This effect is accomplished at least partially through the induction of HSP70, which is part of the next reason; HT causes the release of several heat shock proteins (HSPs) [26,28]. The tumor-specific response will be initiated mostly through the release of HSPs, which can be seen as the most important DAMPs due to HT. HT is also able to induce the other critical DAMPs needed for ICD, such as HMGB1. Lastly, and as discussed before, HT is capable of inducing ROS, another crucial requirement for ICD [12,24,26].

A graphical overview of ICD induction is shown in Figure 3.

#### 1.3.3. Hyperthermia Enhances Immune Cell Trafficking and the Immune Response

Based on the previously discussed mechanisms, it becomes evident that HT can initiate and/or amplify an immune response. In addition, HT exerts direct effects on the immune cell trafficking, making the tumor and its environment more accessible for therapies [7].

Since HT increases the number of immunogenic signals in the form of DAMPs on the surface of cancer cells, the activation of antigen-presenting cells (APC) (e.g., DCs) would be the primary contributing component. In parallel, HT will also repress immunosuppressive tolerogenic signals [12,29]. Through the inhibition of such signals, HT promotes the maturation of several types of APCs, including macrophage polarization to the pro-inflammatory M1 type, and stimulates the release of inflammatory factors [12]. An increase in several cytokines, including but not limited to interleukin (IL)-1β, IL-6, IL-8, IL-10, G-CSF, and tumor necrosis factor alpha (TNF-α) [30,31], have been observed in the peripheral blood within hours after HT treatment.

HT will initiate the maturation of DCs to a CD11c+ population, and the role of HSPs herein is of particular interest [32]. While these HSPs are not inherently immunogenic in nature, they can become so through the formation of a complex with MHC I or II receptors. Such a HSP complex can then bind with APCs through specific surface receptors; for example, HSP70 has been reported to bind monocytes through a CD14 receptor, while gp96 uses CD91 and HSP60 has been shown to be a ligand for TLRs on macrophages. As a direct consequence of this exposure, inflammatory cytokines such as IL-6, IL-12 and TNF-α are secreted and will support DC maturation [31,33,34,35,36]. However, one should note that the HSP family is broad and while some of them, such as HSP70, can be seen as immunostimulatory towards the environment, others like HSP90 assume a more dual role, being both immunostimulatory or protective based on the situation at hand [29,32]. The eventual outcome of applying HT is therefore highly dependent on which HSP will be engaged in response to treatment; in turn, this factor is influenced by the cell type and treatment parameters like temperature [32]. Furthermore, DCs that have been exposed to heat stress may exhibit a distinct expression profile wherein the upregulation of insulin growth factor binding protein 6 (IGFBP-6) should be noted. This protein has been found in recent years to be tightly associated with the immune response through various mechanisms including (but not limited to) improving chemotaxis, activation of pro- as well as anti-inflammatory functions, and involvement in angiogenesis [37,38].

Because of DC maturation under heat stress, the expression of certain surface markers will be increased. Among others, the levels of major histocompatibility complex (MHC II) and costimulatory molecules such as CD80 and CD86 will be elevated, with some studies indicating that MHC I is stimulated as well [12,39,40]. In turn, this enhances the immune response through increased cytokine production and increases in antigen presentation and T-cell activation. The production of cytokines is dependent on the specific immune context and the characteristics of the HT exposure [31,41]. In humans, two distinct types of DCs can be distinguished; both of them will strongly induce the proliferation of CD4+ T-cells. It has been shown that mature DCs will induce Th1 polarization through IL-12 signaling, while it seems that the more immunosuppressive Th2 cells are developed by default; however, it must be noted that this process is more complex and not only IL-12-dependent [42,43,44]. Meanwhile, both activated CD4+ and CD8+ cells will produce cytokines themselves, further helping the differentiation of CD4+ helper cells. Specifically, IL-2, IFN-γ, TNF-α, and GM-CSF will be secreted by Th1 cells, which will stimulate macrophage activity, as well as more MHC I expression on the CD8+ cytotoxic T-cells [31,42]. CD8+ cells, once activated, have no need of co-stimulation to exert their cytotoxic effects; the antigen-bound MHC I complex suffices. Subsequently, these cells will proceed to eradicate the target cells (cancer cells) through the action of three effector molecules: granzyme, perforins, and Fas ligand. These will release other cytotoxic cytokines in the microenvironment, such as IFN-γ, TNF-α, and TNF-β. This can lead to a direct effect on the tumor through macrophage activation, yet depending on the microenvironment, this can also stimulate tumor progression due to the suppression of endothelial adhesion molecules [31,45]. Moreover, HT can have a direct effect on cytotoxic T-cells as well. Thermally treated CD8+ T cells exhibit stronger differentiation towards an effector phenotype with substantially enhanced cytotoxic activity and increased production of interferon (IFN) in both in vitro and in vivo models. Similarly, heat-induced changes in CD4+ cells can lower the requirement of stimulation by CD28 for IL-2 production [46].

In addition to DCs, other components of the innate immune system can be influenced by HT. It has been reported that the expression levels of several macrophage-activating factors, such as chemokine ligands (CXCLs), IL factors, and NO, are increased under HT [29]. Macrophages are affected by heat stimulation via both HSP-related mechanisms, as well as through immunogenic effects [29]. Again, the involvement of HSP70 and HSP90 is of great importance, as HSP70 has been linked to macrophage activation, enhanced phagocytosis, the activation of NF-kB signaling, and inducible nitric oxide synthase (iNOS) expression [47,48]. HSP90 has the same effects but influences NADPH oxidase (NOX) expression instead of iNOS and is therefore also involved in oxidative stress [49,50]. Other HSP family members such as HSP23, HSP27, and HSP40 are involved in macrophage activation in both in vitro and in vivo settings [51,52,53]. However, some studies have observed the exact opposite and reported that induction of the heat shock response inactivates macrophages and decreases their survival when heat shock was strong and persistent. This accentuates once more the importance of fine tuning HT parameters for treatment [29,54,55].

A last component of the innate immune system that can be affected by HT is the natural killer (NK) cells [31,32]. Under HT conditions, it has been observed that there is an HSP70-mediated augmentation of NKG2D receptor clustering, while the total expression on the surface remains the same. Such clustering is also observed when NK cells are directly stimulated to activation via IL-2 [56,57,58]. In short, NK cells will become activated after exposure to HT. Another consequence of heat is the upregulation of MICA on target cells, facilitating the binding of the NK receptors [29,56,58].

To summarize, HT can have an enhancing or activating effect on innate cells, including APCs, which will then prompt the adaptive immune system to get involved. However, to be able to exert an immune response, not only is there a need for activation but infiltration of the activated cells towards the tumor site needs to be facilitated as well.

To this end, HT initiates two reactions at the level of the blood vessels that are of importance. The first one is a rudimentary mechanical effect that can augment perfusion in the tumor. Multiple studies have proven this; for example, a study by Dewhirst and colleagues heated a rat model at 42 °C for 1 h. An increase of 35% in the diameter of arterioles at the tumor-entering site was reported [59]. The dilation and subsequent increase in blood flow facilitate immune cells reaching the tumor site and thus increase their trafficking [32,60].

The second effect happens at the molecular level and involves the expression of chemoattractant molecules and an increase in vessel permeability. This will then cause the immune cells to adhere to and eventually extravasate through the vessel wall. Under regular conditions, a preclinical study by Fisher and collaborators reported an impairment of efficient vessel–lymphocyte interactions in tumor vasculature, thereby hampering normal extravasation. However, after the administration of a mild HT treatment, a 5-fold increase in cytotoxic T-cell homing was observed [58,61]. These findings can be explained by the influence HT has on several steps of the adhesion and migration cascade, which indeed results in increased trafficking of CD8+ lymphocytes across the vessel barrier. In particular, HT enhances tethering and rolling, mediated via mostly L-selectin (as well as some other selectin family members) and increases stable binding via intercellular adhesion molecule-1 (ICAM-1) [12,46,58]. Furthermore, it also augments the binding activity of α4β7 integrin to mucosal addressin cell adhesion molecule 1 (MAdCAM1) at the level of Peyer’s patches in the high endothelial venules (HEV) [62].

The consequence hereof is a reduced rolling velocity of the lymphocytes in this area, which in turn results in increased ability of the HEVs to uphold a stable arrest of lymphocytes. One of the molecules exerting a primary role in this effect is the chemokine ligand (CCL) 21, which can be expressed by the HEVs and is upregulated under HT. The slowed rolling allows for the lymphocytes to inspect the microenvironment on the luminal vessel surface; following the ligation of CCL21 with the CCR7 receptor on a circulating lymphocyte, conformational changes will be induced in the β2 integrin leukocyte-function associated adhesion molecule-1 (LFA-1). Subsequently, LFA-1 can then employ its counter-receptors ICAM-1 and 2 on the HEV surface [46,62]. In addition, the ICAM1 elevation in response to HT also likely promotes diapedesis into the tissues via the formation of adhesive patches [46]. In addition to adhesion molecules, cytokines are important regulators of lymphocyte trafficking as well. The release of several pro-inflammatory cytokines, such as IFN-α, TNF-α, IL-6, and IL-1β, is stimulated by HT [63]. Several studies have reported that especially the tightly regulated IL-6 trans-signaling is pivotal as a trafficking effector. Therefore, this cytokine has been attributed a dual role in cancer progression, since it can be both associated with tumor-promoting mechanisms and at the same time has been observed to exert effects on lymphocyte infiltration [12,46,58,61,63].

The most important impact of all these mechanisms working together is an increased homing ratio of CD8+ T-cells in the tumor microenvironment (TME), while the number of CD4+, CD25, and regulatory (Foxp3+) T-cells decreases. Moreover, this effect is also extremely specific to the tumor site. A causal link has been established between this T-cell homing, the lysis of tumor cells, and an improvement in tumor control [58].

Considering the importance of lymphocyte trafficking in an immune response, it is not surprising that multiple studies have confirmed that the tumor vasculature is crucial in improving immunotherapy efficacy [58]. Mild HT treatment can play a role in this as well; more specifically, it stimulates endothelial cells to express angiopoietin (Angpt) 1 and 2. It also has been reported to induce vascular endothelial growth factor (VEGF) expression, a molecule for which the role in vessel permeability is considered controversial. Consequentially, it has been hypothesized that vasculogenesis and immune cell infiltration are increased due to these factors under febrile conditions [29,64].

#### 1.3.4. Hyperthermia and the Expression of Co-Inhibitory Molecules

Conflicting statements have been reported in the available literature on the topic of co-inhibitory molecule expression under HT. Some studies found that these molecules, such as PD-L1, are downregulated under HT; however, others have reported the opposite [65,66,67,68,69]. Both up- or downregulation could have a significant impact on the treatment strategy. In the case of upregulation, the tumor is able to protect itself more under heat stress through immunosuppressive mechanisms. Therefore, treatment with ICIs could provide larger benefits in such patients by countering this HT effect. On the other hand, if HT would downregulate the expression of co-inhibitory molecules, this implies that, intrinsically, heat already acts as an ICI itself; this would sensitize tumor cells to immune reactions. Further treatment with ICIs could then provide additive or synergistic effects for an enhanced response [12,67,68]. However, to reach a conclusive answer in this matter, more studies are warranted.

#### 1.3.5. Hyperthermia Alleviates Immunosuppressive Hypoxia

Solid malignancies are characterized by their ability to sustain rapid growth under unfavorable conditions. To maintain this abnormal proliferation, tumors require an ample amount of blood to provide them with oxygen and nutrients. Therefore, once the existing vasculature becomes inadequate to support growth, signaling molecules such as VEGF, which promote angiogenesis, are triggered [70,71]. However, newly formed blood vessels in the tumor tend to be abnormal and structurally compromised. They are often leaky, have an irregular shape, and are insufficiently perfused, leading to poor blood flow and oxygenation of the tumor tissue. As a result, regions of hypoxia develop inside the TME [72,73,74].

The TME undergoes several phenotypic alterations as a result of hypoxia, which leads to more aggressive tumor behavior. Among others, hypoxia can suppress an immune response, enabling immune evasion by altering the tumor–immune cell interplay and fostering an immunosuppressive TME. These consequences collectively contribute to tumor progression, metastasis, and therapeutic resistance [72]. Given its crucial involvement in tumor development, hypoxia is an appealing target for therapeutic intervention. Anti-angiogenic pharmaceuticals, hypoxia-activated prodrugs, and oxygenation therapy are some of the methods currently in use to attempt to alter the hypoxic TME. Targeting hypoxia-inducible factors (HIFs) alongside associated signaling pathways also offers intriguing approaches to stop the adaptive reactions brought on by hypoxia. Combination therapies using both hypoxia-targeting strategies and conventional treatments have a great potential to enhance patient outcomes and treatment responsiveness [75,76,77].

As mentioned previously, HT is a therapy that can improve the oxygenation status of a tumor [78]. This can prove important to alleviate the immunosuppressive hypoxic environment, which can hamper or interfere with immunotherapy. Several mechanisms are involved in the effects HT has on hypoxia [58]. Firstly, at the physiological level, heat will activate certain thermoregulatory responses, which, as explained before, will cause the arterioles to dilate. The resulting increase in blood flow and perfusion leads to a decrease in hypoxia, as well as interstitial pressure, in murine models. Consequently, this ensures a less hypoxic environment, which facilitates trafficking of immune cells and therapeutic agents [58,78].

Another important effect is determined through the modulation of HIF-1 expression under HT. HIF-1 is an important transcription factor that plays a crucial role in the response to hypoxia [58,79]. As a master regulator of the cellular response, it is involved in the activation of multiple target genes associated with tumor progression. This factor regulates both the delivery of oxygen and oxygen consumption through its influences on angiogenesis and glycolytic metabolism [80,81]. The most important target genes are VEGF and multiple glycolytic enzymes, such as glucose transporters 1 and 3 (GLUT1 and GLUT3), hexokinase (HK), glyceraldehyde 3-phosphate dehydrogenase (GAPDH), phosphofructokinase-1 (PFK-1), and many others [81]. VEGF, which aids in the proliferation, migration, and tube creation of endothelial cells, is upregulated by HIF-1 expression. Moreover, HIF-1 promotes the synthesis of angiopoietins and platelet-derived growth factor (PDGF), two additional pro-angiogenic factors. In this way, HIF-1 aids in the development of a sufficient blood supply, ensuring that nutrients and oxygen are delivered to hypoxic tissues [82,83].

In addition to providing nutrients and a blood supply to the rapidly growing tissue, HIF-1 also influences the processing of these nutrients. HIF-1 orchestrates metabolic reprogramming to enable cells to adapt to the hypoxic environment. As a result, HIF-1 encourages and provides cancer cells with the necessary energy and building blocks for their rapid proliferation and survival in the harsh TME [84,85].

In most tumor types, HIF-1 is highly expressed due to hypoxia and contributes to a poor prognosis through the previously described mechanisms. Mild HT has been reported to activate HIF-1 and its downstream targets via the ERK pathway through NADPH oxidase-mediated ROS production [79]. Therefore, the effects of HT on HIF-1 could be beneficial or the exact opposite. While the oxygenation status will likely be improved through HIF-1 activation under mild HT, the same upregulated HIF-1 might also contribute to tumor cell resistance and aggressiveness. Therefore, combination with therapies that can counter the injurious effects of HT through HIF-1 could be a viable option to acquire better outcomes. Once more, this also accentuates the need for therapy planning when combining therapy with HT-based treatments, as the timing can be of crucial importance in counteracting the detrimental effects of HT while enhancing the beneficial ones. In short, HT can clearly provide a benefit against hypoxia, thereby rendering the microenvironment less immunosuppressive. At the same time, it should be used with caution because of its cellular effects on HIF-1 [58,79].

The objective of this review is to investigate the literature on the effect of HT in combination with emerging therapies such as ITs and TTs as potential cancer treatments. We hypothesize that HT as a sensitizer could improve the working mechanisms of several compounds, and thereby improve the potential as a successful combination therapy in cancer treatment (Figure 4).

## 2. Materials and Methods

### 2.1. Pipeline Research

For the determination of potential targets, the oncology pipelines of 15 large pharmaceutical companies were reviewed (Table 1) between May and September 2022. Two independent researchers extracted the compound information from the available data; this included the compound name, study phase, cancer type(s), mechanism(s) of action, therapy type(s), and target(s). For some companies, no data was available on phase I and/or IV stage compounds. An overview of the research strategy is summarized in Figure 5. A full list of all compounds can be found in the Appendix A.

### 2.2. Identification of Potential Synergism

An initial preliminary search was then started to identify molecules with a potential synergy with HT. This was performed using systematic PubMed queries including the target or compound name(s) in combination with the MeSH term “Hyperthermia”. If ≥ one publication(s) were found that linked the target together with HT, the compound was labelled as a potential synergistic molecule.

### 2.3. Target Characterization and Selection

From the generated list of compounds, 145 unique targets were identified, many of which also appeared in combinations. Of those unique targets, the compounds that were of most interest based on the preliminary synergy search and their number of occurrences in combinations or over multiple pipelines were identified. This way, the decision was made on three targets of interest to further explore in this review. This includes two types of IT, the ICI anti-programmed death protein 1/protein ligand 1 (PD-1/L1) and BiTEs in general (based on the multitude of CD-related compounds in the pipelines). One type of TT based on VEGF was also included due to its interesting features in combination with HT.

## 3. Results

In this manuscript, three potential targets are discussed, including PD1-1/L1s, BiTEs (CD-related compounds), and VEGFs.

### 3.1. PD-1 and PD-L1

The PD-1/PD-L1 signaling axis is one of the main immune checkpoints that helps regulate the threshold in antigen responses of both T- and B-cells (Figure 6) [86]. PD-1 in particular is highly expressed in tumor-specific T-cells and its expression can be upregulated under certain conditions, such as a chronic infection or cancer cell leakage [87,88,89].

This molecule is associated with two ligands, namely PD-L1 and PD-L2 [90]. PD-L1 is the more prominent one of the two. Due to its upregulation in T-cells, B-cells, epithelial cells, and endothelial cells after the expression of certain cytokines (e.g., IFN-γ), this protein is considered to help safeguard peripheral tolerance [87,90,91].

A key factor in immune response initiation and regulation is the activation of T-cells. Such a response is achieved via the two-signal model, wherein the first step relies on antigen recognition via T-cell receptor (TCR) and MHC interactions. The second signal, however, is independent of the antigen itself and is considered co-stimulatory or co-inhibitory [86,91,92]. Anergy of the T-cell occurs in the absence of this second stimulus, thereby aiding in the preservation of self-tolerance but also tolerating a poor anti-tumor response [87,91]. PD-1 and, by extension, its ligand PD-L1 are co-inhibitory molecules: when they bind with each other, the T-cell response will be interrupted at the stage of the second signal and therefore be aborted. The working mechanism of PD-1/L1 is shown in Figure 6.

#### 3.1.1. PD-1 and PD-L1 in Cancer and the Tumor Microenvironment

##### The Role of IFN-γ

PD-1 and PD-L1 mechanisms play a crucial role as regulatory pathways in controlling the immune system to prevent excessive activation [93]. Tumor cells must develop strategies to evade immunosurveillance. To accomplish this, they can undergo phenotypic changes under immune-induced stress, also known as immunoediting. Herein, the PD-1/PD-L1 axis is a target of interest due to its intrinsic involvement in immune responses [94,95,96].

PD-L1 is not expressed on most normal cells, but its expression can be induced in almost any nucleated cell. An important cytokine involved in its induction is the inflammatory cytokine IFN-γ, a molecule originating mostly from activated T-cells but also other (activated) immune cells [97]. IFN-γ has several mechanisms through which it can regulate PD-L1 expression, dependent on the type of tumor; in most cases it involves JAK-STAT-related pathways and the PI3K/Akt pathway, while the MEK/ERK and NF-Kβ signaling pathways are also known to be implicated in some tumors [97,98,99,100,101,102]. Therefore, when the immune system attempts to eliminate the tumor cell and thereby activates those cells, this will indirectly generate IFN-γ and in that way lead to the expression of PD-L1 on the tumor cells themselves [97]. Considering that all activated T-cells are already expressing the PD-1 receptor, this can induce a paradoxical situation where the immune system is in fact actively helping the tumor cells as a consequence of rendering itself unable to now activate a response [91,96].

It must be noted that in a subfraction of tumors, PD-L1 is already constitutively expressed without IFN-γ stimulation due to a lack of activated TILs. This can be attributed to the involvement of oncogenic signaling pathways with a deletion or silencing mutation of PTEN, ALK, or EGFR. Although quite rare, an intrinsic oncogenic induction of PD-L1 has also been observed in some cases [91,103,104].

##### Other Cellular Factors Involved in PD-L1 Induction

The cytokine TNF-α is key factor in the TME that can contribute to the modulation of PD-L1 expression. Contradictive to its usual inflammatory function, it has been found to upregulate PD-L1 expression via NF-Kβ and ERK1/2 pathways [97,105]. However, some publications show that it can also suppress PD-L1 via miRNA-155, contributing to its conflicting role [97,106].

Several other cytokines are also able to regulate PD-L1 expression. A few of the more important players related to this are IL-6 [97,107,108,109,110,111], IL-10 [97,111,112,113], IL-12 [114], IL-17 [105], and IL-27 [115].

Yet another mechanism to upregulate PD-L1, frequently seen in types of lung cancer (but not limited to this), builds on a mutation in the EGFR pathway. Recent studies have shown that not only does such a mutation promote malignant proliferation and metastasis but it also aids in immune escape. Per cancer type, the specific mechanism to achieve this, however, differs greatly [97,104,111,116].

##### Hypoxia and PD-L1

Especially in solid tumors, an important element in development and progression is the hypoxic environment. Due to the high proliferation rate of cancerous cells, the formation of well-functioning blood vessels is severely under stress. As this hampers the blood supply to the tumor, the oxygen content and amount of nutrients available for the growing cells become limited [81,97,111]. Therefore, tumor cells start to metabolically reprogram themselves to adapt to these harsh conditions. In order to survive the hypoxic environment, immune escape also remains an important strategy [97].

The most important proteins involved are part of the HIF family. Under normoxia (normal oxygen levels), HIF subunits are hydroxylated, causing them to be identified for ubiquitination and degradation. Under hypoxic stress, this hydroxylation is inhibited and the HIFα protein levels stabilize [81,97]. The result of this is that via its target pathways, HIF will now promote tumor immune escape, mainly via PD-L1 induction. It has been observed that in such hypoxic conditions, the expression of PD-L1 is elevated in several types of immune cells (T-cells, DCs, and others) and cancer cells [117,118]. Experiments have confirmed that this effect is transcriptionally regulated via HIF-1α or HIF-2α binding at the PD-L1 HRE promoter. Moreover, HIF-1α and HIF-2α can directly upregulate PD-1 as well. It has also been noted that PD-L1 increases glycolysis via its downstream effectors Akt and mTOR, taking away glucose from CD8^+^ T-cells [81,97,118].

The list of mechanisms controlling PD-L1 expression described above is limited to the most important contributors. A variety of epigenetic, post-transcriptional, and post-translational modifications can have an impact as well [97,111].

#### 3.1.2. Current Therapies

To overcome the “adaptive resistance” of tumor cells to treatment, there has been a notable increase in the development of compounds that target the PD-1/PD-L1 pathway, both as single-agent treatments and in combination with other drugs. Currently, several compounds have already been approved by the FDA for clinical use, with dostarlimab being the latest one gaining its first approval in 2021 [119].

In the category of PD-1 antibodies, molecules such as nivolumab, pembrolizumab, cemiplimab, and dostarlimab can be found. PD-L1 antibodies include atezolizumab, durvalumab, and avelumab. The first approvals for the PD-1 antibodies pembrolizumab and nivolumab were given in 2014 as a second-line treatment for melanoma. In the following years, these immunotherapies received approval for non-small cell lung carcinoma (NSCLC), renal cell carcinoma, and other solid cancers. From 2017 onwards, the repertoire of PD-1/L1 antibodies as well as cancer types that they can be used for has been rapidly expanding [93,119].

#### 3.1.3. Benefit from Combination with Hyperthermia

Although ICIs are shown to be promising in many trials, there is also a substantial subgroup of patients for whom this treatment fails to yield a significant response [120]. An explanation for these varying outcomes among patient groups is that ICIs such as anti-PD-1/L1 antibodies provide a better response when given to a “hot” tumor rather than a “cold” one (Figure 7) [12]. A hot tumor can best be described as immune-infiltrated, inflamed, and activated due to the presence of CD8^+^ cytotoxic lymphocytes and other TILs at the tumor center and invasive margins. Additionally, this may include the expression of PD-L1 on tumor-associated immune cells, genomic instability, the presence of inflammatory cytokines or chemokines, or evidence of a pre-existing antitumor response [65,121,122,123]. HT [120], in turn, is able to achieve such an inflamed/hot tumor environment via several mechanisms that activate immune modulatory effects and, subsequently, immune system activation (see the previous section). Therefore, treating tumors with HT and ICI-based therapies might induce a more favorable response than when using ICIs alone.

### 3.2. Bispecific Antibodies and Immune Cell Engagers

#### 3.2.1. CD Antigens

The cluster of differentiation (CD) nomenclature was established during the first International Workshop and Conference on Human Leukocyte Differentiation Antigens (HLDA) in 1982. Importantly, the identification of cell surface CD antigens on leukocytes and other cell types has provided new targets for immunophenotyping according to the cell subtypes and stages of differentiation, thus creating new perspectives for researchers [124]. Some CD antigens have also been studied as targets to modulate the immune response due to their restricted expression and roles in immune effector subtypes. In this context, they have been exploited to develop new therapies in oncology.

#### 3.2.2. Bispecific Antibodies and Immune Cell Engagement

The rise of immunotherapy and biotechnology over the past decades has driven bispecific antibodies (bsAbs) from conceptualization to development. BsAbs are recombinant antibody-based constructs distinct from mAbs since they bind two different and specific epitopes instead of one [125,126]. Generally, bsAbs are classified either as IgG-like or non-IgG-like antibodies, which can be distinguished by the presence or absence of the fragment crystallizable (Fc) region [126].

The Fc region is involved in extending the half-life of antibodies by recycling IgG when it binds to the neonatal Fc receptor (FcRn). The presence of the Fc region binding FcγR is also essential for the lysis of opsonized pathogens via antibody-dependent cell-mediated toxicity (ADCC) and complement-dependent cytotoxicity (CDC) [127]. However, while the presence of the Fc region is crucial to mediate the cytotoxic activity of mAbs targeting tumor cells, it is undesirable when the strategy is to engage an immune cell on a tumoral one.

BsAb constructs without the Fc region can thus be designed to avoid ADCC and CDC, but this drastically reduces their size and half-life [127]. The smaller building blocks used to generate bsAb constructs are the single-chain variable fragments (scFv), which are composed of the binding domains (V_L_ and C_L_) of the fragment antigen-binding (Fab) region linked by a flexible peptide. Connecting two different scFv with a peptide chain leads to scFv-based bsAbs. Lacking the Fc region, scFv-based bsAbs are much smaller in size, leading to higher tissue penetration [128]. However, being smaller and unable to be recycled by FcRn leads to rapid clearance from the blood circulation by the kidney. For this reason, they must be continuously administered to the patient for the duration of treatment [129,130].

#### 3.2.3. Mechanisms of Action of Bispecific Antibodies

The mechanism by which a bsAb functions relies on the specific epitopes targeted by each binding site, and can encompass the following: (1) inhibition of signaling pathways by binding two different epitopes [131]; (2) delivery of pre-incubated payloads to targeted cells [132]; and (3) engagement of effector immune cells on tumor cells [133,134]. Most bsAbs currently evaluated in clinical trials fall into the latter category. The mechanism consists of targeting both a CD antigen specifically expressed by an effector immune cell type and an antigen highly specific of the tumor otherwise known as TAA [135]. The interaction between the CD antigen and the TAA then trigger the activation of the immune cell and thereby a cytolytic response against the tumor cell. Interestingly, TAAs are not required to be involved in pathogenesis to be considered a target of interest; the aim here is to induce specific immune cell-mediated lysis of tumor cells while sparing healthy cells. Activated effector cells then proliferate, which amplifies the response. For now, immune cell engagement can involve phagocytic, NK, and T-cells since these cells can be engaged by binding specific CD antigens, such as CD64, CD16 and CD3, respectively [136,137,138]. However, only T-cell engagers have so far demonstrated clinical benefits leading to their approval.

Among the various constructs that engage T-cells, most are BiTEs, a type of non-IgG construct composed of two scFv fragments binding both CD3 and a TAA. CD3 is a co-stimulatory receptor of the TCR commonly expressed on the surface of T-cells, regardless of their subtypes. CD3 is indispensable for the full activation and proliferation of T-cells since its major function is to transduce the signal induced by the interaction of the TCR with the antigenic peptide presented by MHC-I [139]. CD3 has been targeted in both inflammatory diseases and malignancies; thus, agonist and antagonist anti-CD3 antibodies have been developed to induce a tolerogenic or immunogenic response, respectively. For BiTEs, a cytolytic response is induced by binding both CD3 on the surface of a cytotoxic T-cell (CTL) and a TAA. Firstly, the BiTE binds its targets and creates a bridge to bring the CD3^+^ T-cell and the TAA^+^ tumor cell into close contact. The proximity between the CTL and the tumor cell induces the formation of a cytolytic synapse and CTL activation independently of the TCR/MHC-I interaction. By bypassing the TCR/MHC-I interaction, BiTEs allow a common mechanism of immune escape found in many cancers to be circumvented that consists of MHC-I downregulation on the surface of tumor cells [140,141,142]. Following activation, cytotoxic T lymphocytes (CTLs) release vesicles that contain perforin and granzymes. When perforin is released in the presence of extracellular calcium, it forms pores in the membrane of tumor cells, enabling the entry of granzymes. These granzymes then cause damage to the targeted cells. Granzyme B plays a significant role in inducing apoptosis in tumor cells. It functions as a caspase by cleaving substrate proteins, including pro-caspases 3 and 7 [143]. As a result, granzyme B and activated caspases lead to membrane blebbing, DNA damage, and eventually the lysis of tumor cells [144].

In parallel, CTLs activated by BiTEs enhance the immune response by releasing pro-inflammatory cytokines such as IL-2, IFN-γ, and TNF-α [145,146]. Ross et al. showed that BiTEs can trigger the lysis of tumor cells not expressing TAAs in an in vitro solid tumor model. Indeed, an EGFR × CD3 BiTE was able to induce the bystander effect by killing EGFR^−^ tumor cells when co-cultured with EGFR^+^ cells. The mechanism involved IFN-γ and TNF-α released by BiTE-activated T-cells, and the upregulation of ICAM-1 and FAS at the surface of TAA^−^ cells [147]. These results are particularly interesting for solid tumors since most of them have a strong heterogeneity of TAAs [148]. BiTEs can also redirect CD4^+^CD25^+^ Treg cells to tumor cells since these lymphocytes express CD3. Despite their role in mediating a tolerogenic response, Treg cells express perforin and granzyme necessary for tumor cell lysis [149]. However, these results have been mitigated by other studies showing that Treg cells activated by BiTEs release IL-10, mediating the inactivation and apoptosis of CTLs [150].

#### 3.2.4. T-Cell Engagement in Hematological Malignancies

The first BiTE, blinatumomab, was approved by the FDA in 2014 for relapsed or refractory (R/R) CD19^+^ B-cell precursor acute lymphoblastic leukemia (ALL). Blinatumomab engages both CD3 and CD19, a costimulatory receptor of the BCR mostly expressed at the surface of B-cells, regardless of the stage of differentiation. After blinatumomab binds its two targets, engaged T-cells lyse normal and malignant CD19^+^ B-cells. In the meantime of CD19^+^ B-cell depletion, healthy B-cells are continuously restored by hematopoietic stem cells [151]. In a clinical trial including 405 adult patients, blinatumomab significantly improved the management of patients (median overall survival of 7.7 months vs. 4.0 months in the CT group and complete remission of 42% vs. 20%). These results have provided a strong rationale to further investigate bsAbs in hematological malignancies [152,153].

BCMA and CD20 are the two other most investigated targets in the development of bsAb for hematological malignancies [125]. Teclistamab and elnaramab, two BCMA × CD3 bsAbs received conditional approval from the Food and Drug Administration (FDA) and European Medicines Agency (EMA) for the treatment of patients with relapsed or refractory multiple myeloma. The effect of this bsAb on BCMA^+^ relapsed/refractory multiple myeloma was compared both in vitro and in vivo to anti-BCMA CAR-T cells [154]. REGN5458 treatment of humanized NOD-SCID-IL2Rg^null^ mice induced a T-cell-dependent immune response in the bone marrow TME, leading to cytokine production, T-cell expansion, and then tumor clearance. Interestingly, the antitumor activity was similar but faster than the one observed with anti-BCMA CAR-T cells.

CD20 × CD3 BsAbs have also been increasingly investigated in hematological malignancies. CD20 is a non-glycosylated phosphoprotein belonging to the membrane-spanning 4-domain family A (MS4A) expressed at the surface of normal and malignant B-cells that plays a role in proliferation and differentiation [155]. Even though its role is still unclear, it has been reported that CD20 is involved in B-cell activation and the humoral response [156,157]. CD20 × CD3 bsAbs showed robust results, leading to the approval of mosunetuzumab and glofitamab for adult patients with relapsed/refractory follicular or large B-cell lymphoma.

Despite the significant improvement in patients with hematological malignancies, some resistance mechanisms involved in therapeutic failure or relapse have been described. Loss of TAAs on the B-cell surface is strongly implicated in relapse after bsAb treatment. After the administration of blinatumomab, cellular phenotypes were not altered except for CD19. Among CD19 alterations, splice variants impeding the recognition of TAA-derived epitopes or anchorage at the surface of the B-cells were observed [158]. This supports the hypothesis of isolated mutational events driving ALL relapse. Another mechanism involves a lineage switch in patients with KMT2A (lysine methyltransferase A2) rearrangements in mixed lineage leukemia-rearranged ALL [159,160]. Different expression patterns have been observed, depending on the targeted TAA. For instance, CD22 targeting strategies showed similar results to immunotherapies targeting CD19 with 70% remission in patients with R/R ALL [161]. Therefore, engaging CD22 appeared as a strategy in case of non-response or relapse next to therapies targeting CD19. However, CD22 expression is more variable than CD19 expression in ALL, and patients treated with anti-CD22 immunotherapies showed a higher rate of relapse. Here, resistance to bsAbs targeting CD22 involved the regulation of CD22 expression via the inclusion or skipping of exon 2 [162]. Interestingly, Ramakrishna et al. showed that the administration of Bryostatin-1, a molecule upregulating CD22 on the B-cell surface was able to improve CD22 persistence and response both in vitro and in vivo [163]. These results suggest that the combination of bsAbs with agents that upregulate or stabilize TAA expression on the surface of tumor cells is a promising strategy to counteract resistance mechanisms.

#### 3.2.5. Challenges for T-Cell Engagement in Solid Tumors

After having demonstrated a clinical benefit in hematological cancers, bsAbs have been increasingly studied in solid tumors. The first approved T-cell engager for solid tumors was catumaxomab. As previously described, this EpCAM × CD3 IgG-like bsAb was approved in 2009 for the treatment of malignant ascites and withdrawn from the market for commercial reasons [164,165,166,167,168,169]. Many different EpCAM × CD3 T-cell engagers have been developed and have shown promising results in preclinical studies [170,171,172,173,174,175,176]. Some of them are currently being evaluated in clinical trials [177,178,179]. So far, the most extensively studied TAAs in solid tumors are carcinoembryonic antigen (CEA) [180,181,182,183,184,185], prostate-specific membrane antigen (PSMA) [186,187,188,189,190,191,192,193,194], HER2 [195,196,197,198,199], and EGFR [200,201,202,203,204,205,206,207,208].

Despite growing clinical research, T-cell engagers have exhibited limited efficacy in treating solid tumors. One significant challenge is the occurrence of on-target, off-tumor toxicity. Currently, most TAAs investigated in solid tumors are expressed at higher levels in specific tumor types, but they are also found at lower levels in healthy tissues. To address the issue of off-tumor targeting, a strategy involves the development of bsAbs that specifically target tumor-specific antigens (TSAs). TSAs are antigens expressed only in tumor tissues; multiple mechanisms of TSA generation have been described, such as alternative splicing, single nucleotide variation, gene fusion, and insertion or deletion [209]. More recently, tebentafusp, a gp100 peptide-HLA-A2 × CD3 T-cell engager, has been approved both in the USA and Europe for the treatment of uveal melanoma and malignant melanoma, opening new perspectives in TSA-targeting approaches [210].

The physical barrier created by extracellular matrix stiffness is another factor reducing the efficacy of bsAbs in solid tumors by impeding the diffusion and accumulation of both drugs and immune effectors [211,212].

The populations of immunosuppressive cells found in immunologically “cold” tumors promote a tolerogenic environment via the release of cytokines and molecules (e.g., IL-10, TGF-β, indoleamine 2,3-dioxygenase (IDO) [213], and CXCL12 [214]) inhibiting tumor cell clearance by immune effectors. Unfortunately, the engagement of Treg cells is another factor involved in resistance to T-cell engagers because of the release of cytokines inhibiting T-cell activation.

#### 3.2.6. Potential Benefits in Combination with Hyperthermia

Some HT-mediated mechanisms have an impact on key factors involved in the response to T-cell engagers; they include (1) the number of T-cells infiltrating the tumor site [215]; (2) their quality (pro-inflammatory CTLs or pro-tolerogenic Tregs) [150,215]; and (3) an environment promoting their activation, which depends on factors such as the TAA density or released cytokines [216,217].

As previously described, the ICD triggered by HT promotes DAMP expression on the tumor cell surface, helping the maturation of several types of APCs that release inflammatory cytokines and chemokines [12,30,31]. By promoting APC maturation and so the release of pro-inflammatory cytokines, HT promotes the differentiation of T-cells into CTLs instead of Treg cells. Furthermore, it was shown that HT promotes the Th1 phenotype and decreases Foxp3 expression, a marker of Treg cells, on the T-cell surface [218]. This effect involved Foxp3 degradation via a ubiquitin-proteasome pathway [219]. Taken together, these results are particularly interesting since it has been shown that an increased percentage of Treg cells is associated with a reduced response or relapse after treatment with blinatumomab [150].

HT can also promote T-cell activation. Indeed, it has been shown that HT in combination with CT resulted in a doubling of the expression of CD69, a marker of T-cell activation. In addition, increased concentrations of TNF-α and IFN-γ were observed. However, in this study, no effect was observed with HT alone [220]. Similarly, Mace et al. showed that HT was able to enhance antigen-dependent effector T-cell activity. This higher activity was correlated with LAT and PKC activation and was related to increased production of IFN-γ [221].

Since no study describes the combination of a T-cell engager with HT, it is needed to further investigate these results to come to a conclusion.

HT holds promise in enhancing drug delivery and facilitating the diffusion of effector immune cells within the tumor site. This characteristic is particularly valuable for solid tumors with limited perfusion. Under mild HT conditions (40–42 °C), an increase in blood flow and vascular permeability has been observed. This effect enhances the accessibility of the tumor and its surrounding environment to therapeutic interventions, while also promoting the trafficking of immune cells to the site [222].

Finally, HT can induce the accumulation of DNA damage in tumor cells, leading to the development of neoantigens or TSAs that are unique to the tumor and not present in healthy cells [19,20,21]. The mechanisms causing the generation of these neoantigens/TSAs through HT involve various types of somatic mutations, such as amino acid changes, frameshift mutations, and insertions or deletions [223]. This discovery holds great potential in the field of personalized medicine, offering new perspectives for tailored therapeutic approaches.

### 3.3. Vascular Endothelial Growth Factor (VEGF)

#### 3.3.1. Vascular Endothelial Growth Factor Biology and Function

Ferrara et al. (1989) first isolated and described the vascular endothelial growth factor (VEGF) protein [224]. In mammals, VEGF ligands consist of five related glycoproteins. So far, VEGFA, VEGFB, VEGFC, VEGFD, and PLGF (placental growth factor) have been identified. These ligands are secreted to form monodimers, which interact with three receptor tyrosine kinases (RTKs): VEGF receptors −1, −2, and −3 (VEGFR1-3) (Figure 8A) [225].VEGFs can be spliced alternatively to produce isoforms with various biological functions [226]. This is due to the fact that these functions are dictated by the ability to interact with VEGFR co-receptors (e.g., neuropilins and heparan sulfate proteoglycans) (Figure 8B).

Next to splicing, proteolytic processing also controls the bioactivity of the VEGF family. This mechanism allows interactions with different receptor types. Each VEGFR gene encodes a protein composed of seven extracellular immunoglobulin (Ig)-like domains, a short transmembrane-spanning polypeptide, and an intracellular portion containing tyrosine kinase enzymatic activity [227]. VEGFRs are particularly unique in their ability to transduce signals that form the three-dimensional vascular tube and regulate vascular permeability [226].

The VEGFRs are found on a wide variety of cell types. VEGFR1 is mainly found on vascular endothelial cells, hematopoietic stem cells, macrophages, and monocytes and is a positive regulator of monocyte and macrophage migration. Furthermore, VEGFR1 has been described as a positive and negative regulator of VEGFR2. VEGFR2, which is expressed on vascular and lymphatic endothelial cells, is implicated in aspects of normal and pathological vascular endothelial cell biology. Last, VEGFR3 has been proven important for lymphatic endothelial cell development and function. VEGFA is the major ligand of the VEGF family and binds to two RTKs, VEGFR1 and VEGFR2 [228].

The VEGF family is a crucial regulator of vascular development during embryogenesis (vasculogenesis), as well as blood vessel formation (angiogenesis) [228,229]. While vasculogenesis exclusively occurs during embryonic development, angiogenesis also occurs during female reproduction, wound healing, and following exercise. Therefore, VEGF-mediated signaling pathways are associated with angiogenetic regulation by influencing endothelial proliferation, migration, survival, extracellular matrix degradation, and cell permeability [230].

The immune system is also affected by VEGF in a variety of ways. Studies in mice have shown that VEGF directly disrupts the maturation of T-cells from early hematopoietic progenitor cells, which impairs DC functions. Moreover, VEGF has been demonstrated to mediate monocyte migration, mobilize circulating endothelial precursor cells, and hematopoietic stem cells [231].

#### 3.3.2. Regulation of VEGF and VEGFR Expression

Dysregulation of angiogenesis may contribute to the progression of various diseases, including cancer, macular degeneration, and diabetic retinopathy [229]. Carmeliet (2005) stated that without an adequate vascular supply, solid tumors can only reach the size of 1–2 mm (around 10^6^ cells), mainly due to a deficiency in oxygen and nutrients [232].

By binding to VEGFR2 to increase endothelial cell proliferation via the RAS-RAF-MAPK-ERK signaling pathway, VEGFA appears to be the major mediator of tumor angiogenesis [233]. VEGFA mainly signals through VEGFR2 [230]. VEGFR2 is expressed at elevated levels by endothelial cells engaged in angiogenesis and by circulating bone marrow-derived endothelial progenitor cells. VEGFA triggers endothelial cell migration, an integral component of angiogenesis [234,235,236].

VEGFA gene expression can be upregulated by a variety of factors, including PDGF, fibroblast growth factor (FGF), epidermal growth factor, TNF, TGF-β, and IL-1 [232].

#### 3.3.3. VEGF Biology in Cancer

VEGF, which is released by tumor cells and the surrounding stroma, increases endothelial cell proliferation and survival, resulting in the development of new blood vessels, which may be architecturally aberrant and leaky [237,238]. The majority of human malignancies contain high levels of the VEGF mRNA, which corresponds with invasiveness, vascular density, metastasis, recurrence, and prognosis [233].

Preclinical in vitro experiments showed elevated VEGF mRNA levels by roughly 10- to 50-fold in response to lowering oxygen levels from 21% to 0–3%. Hypoxic induction of VEGF appears to be a common response. Similar results were seen in vivo [239].

When solid tumors grow in size, cells within the expanding mass generally become hypoxic due to the increasing distance to the nearest blood vessels. VEGFA expression via HIF-1α and HIF-2α was found to be maximal in necrotic regions and regions with a lack of vascular access [239]. HIF is hydroxylated by a class of oxygen- and iron-dependent enzymes known as HIF prolyl hydroxylases under normoxic circumstances, resulting in HIF identification by the von-Hippel Lindau (VHL) tumor suppressor protein. As a result, HIF is polyubiquitylated and thereby capable of destroying HIF-1α. Degradation prevents HIF-1α dimerization and binding to the promotor of the VEGF gene. This, in turn, inhibits the transcription of the VEGF gene and synthesis of the VEGF protein. VHL does not ubiquitinate or degrade HIF-1α in hypoxic conditions. As a result, VEGF synthesis and angiogenesis are promoted by HIF-1α dimerization and binding to the VEGF gene promotor [240]. This supports the idea that angiogenesis is driven by hypoxia, one of the main characteristics of solid tumors [76,241].

The loss of tumor suppressor genes (e.g., p53), as well as the activation of oncogenes, including Kras, Hras, v-src, HER2, HER1/EGF receptor (EGFR), FOS, trkB, v-p3K, PTTG1, and Bcl-2, are related to increased VEGF expression [242]. It has been shown that a number of growth factors increase the expression of the VEGF gene, including PDGF, FGF, TNF, and IL-1. Tumor-derived growth factors encourage tumor angiogenesis in this manner. In vitro experiments also revealed that some tumor cell lines may express VEGF and VEGFR, allowing VEGF to function as both an autocrine and paracrine factor, which results in a positive feedback loop that directly affects tumor cells [243]. The induction of anti-apoptotic proteins Bcl-2 and survivin, both of which are introduced by VEGF, also play a crucial role in tumor progression by protecting the neovasculature of tumors from apoptosis [244]. Additionally, VEGF promotes the release and activation of extracellular matrix-degrading enzymes such as plasminogen activator and the MMP interstitial collagenase, enabling the unrestricted growth of newly formed blood vessels [245]. VEGF is, therefore, a key component in the formation of tumors since tumor angiogenesis and the preservation of the tumor vasculature are crucial for the progression of cancer.

#### 3.3.4. VEGF Inhibitors and Dual Targeting Therapies for Tumor Therapy

VEGFR2 has been identified as a potential promising tumor therapeutic target [246,247]. Accumulating evidence shows that abnormal VEGFR2 expression in neovascular tumor endothelial cells is closely linked to the genesis and progression of multiple tumor types. VEGFR2 inhibitors show a variety of therapeutic efficacies against different malignancies by inhibiting angiogenesis and lymph angiogenesis; however, most of them lack specificity.

Multiple strategies for the inhibition of the VEGF-VEGFR signaling system have been developed for cancer treatment. The discovery of VEGF led to a major step forward in understanding angiogenic pathways; in 1993, Kim and colleagues identified mAbs that were able to target and neutralize VEGFA, thereby inhibiting tumor growth in preclinical studies [248]. This resulted in the development of the recombinant humanized VEGFA-specific mAb bevacizumab, which was approved by the US FDA in 2004 for the first-line treatment of metastatic colorectal cancer, followed by the approval from the EMA and other regulatory authorities [249]. This accomplishment resulted in the continual discovery of several angiogenic inhibitors to treat cancer (Table 2).

After the confirmation that bevacizumab monotherapy was well tolerated with no grade III or IV safety issues, further studies revealed that the addition of bevacizumab to standard-of-care chemotherapies could result in clinical benefits in a variety of tumor types, without increased toxicity [250]. The idea behind combination therapy was to target both endothelial and tumor cells at the same time. Preclinical studies confirmed synergistic effects between bevacizumab and cytotoxic therapies, since VEGFA blockage appears to sensitize the endothelium to the effects of the cytotoxic agents [251,252]. Additionally, VEGFA inhibition causes apoptosis in endothelial cells not covered by pericytes, thereby reducing the abnormal tortuosity and hyperpermeability of the tumor vasculature, causing normalization, and thereby reducing the tumor interstitial pressure and improving the delivery of cytotoxic agents [253,254]. In a pivotal phase III trial in 2004, bevacizumab was combined with irinotecan and 5-fluorouracil and leucovorin; this significantly increased treatment response rates, progression-free survival, and overall survival in previously untreated patients with metastatic colorectal cancer compared to the combination of chemotherapeutic agents alone [249]. As a result, the first VEGF mAb was approved by the FDA in 2004 for the first-line treatment of metastatic colorectal cancer. The addition of bevacizumab to conventional chemotherapies resulted in even more significant clinical benefits in various advanced cancers [255].

In addition, small-molecule VEGFR2 inhibitors, initially described in 1996, have been explored to block the VEGFA-VEGFR pathway for cancer treatment [256]. These early-generation tyrphostin family compounds inhibited VEGFA-dependent VEGFR2 autophosphorylation, as well as various biological functions of VEGFA [257]. Other families of small-molecule VEGFR receptor Tyr kinase (RTK) inhibitors could be developed after the discovery of the crystal structure of the VEGFR2 kinase domain [258]. In addition to VEGFRs, these compounds potentially inhibit other structurally related RTKs (e.g., PDGF receptors, cKIT, FLT3, and macrophage colony-stimulating factor 1 receptor). Some of these small compounds can also block RTKs with other structures (e.g., EGFR, TIE2, cMET, RET, and fibroblast growth factor receptors). As a result, the antitumor action of these compounds may reflect the contributions of many targets in the microenvironment, as well as direct impacts on tumor cell proliferation in some situations.

Two protein inhibitors of the VEGFA pathway—aflibercept, a recombinant VEGFR fusion protein that binds to and inhibits VEGFA, VEGFB, and PIGF21, and ramucirumab, a fully human monoclonal antibody that inhibits VEGFR2—have been approved for use in cancer therapy in addition to bevacizumab and small-molecule RTKIs [259].

#### 3.3.5. Challenges in the Development and Use of VEGF-A Inhibitors in Oncology

Despite the validated VEGF-A inhibitors and some considerable benefits in patients with advanced cancer and limited treatment options, it remains unclear why some patients with specific tumor types show limited responses. This reflects the complexity of a process like angiogenesis, which is regulated by multiple factors in the microenvironment [260]. Drug resistance becomes apparent by the fact that many patients show progressive disease despite anti-VEGFA therapy. The mechanisms involved appear to be fundamentally distinct from those that typically occurring after treatment with inhibitors of well-defined oncogenic pathways that render a drug ineffective (selection present or acquired mutations in the target or in the pathway) [261,262]. There is currently no compelling evidence to support the hypothesis that therapy resistance is caused by mutations in VEGF-A or its receptors.

#### 3.3.6. Hyperthermia and Anti-Angiogenic Therapy

The impact of local HT on VEGF and HIF-1 levels was investigated by Moon et al. [79]. After the application of 1 h of HT at 42 °C, HIF-1 was increased. Through the ERK pathway, NADPH oxidase was upregulated in tumor cells. These findings confirm the idea that local HT alone could promote tumor angiogenesis [79]. Consistent with this hypothesis was the study by Nie et al., where a de novo VEGFR2 inhibitor was combined with HT (40 min at 42 °C) in vitro in human umbilical vein endothelial cells (HUVECs), murine mammary cancer 4T1 cells, and murine colon carcinoma cells. Combination therapy significantly inhibited the proliferation of all cell lines. The 4T1 and CT26 cell lines were engrafted in BALB/c mice and used to examine the combination therapy in vivo. In both models, the combination therapy resulted in inhibited tumor growth and a prolonged life span. Moreover, angiogenesis and increased tumor apoptosis and necrosis were observed in the combination group. In addition, combining therapies could prevent the tumor from metastasizing to the lungs. No additional toxicity was seen in major organs, including the heart, liver, spleen, lung, and kidney [263].

Another study combining HT with anti-angiogenic drugs was conducted by Kanamori et al. SCC VII tumors in C3H/He mice were evaluated by hematoxylin-eosin and immunohistochemical (IHC) staining for VEGF. The % necrotic area of untreated SCC VII tumors was 7%, while tumors treated with anti-angiogenic therapy alone and HT alone were 27 or 65%, respectively. When HT and anti-angiogenic therapy were combined, a necrotic area of 82% was measured, which was significantly higher (*p* < 0.05) than that caused by either therapy alone. Strong IHC staining for VEGF was observed in untreated EMT-6 tumors of BALB/c mice, which have spontaneous central necrosis, whereas weak IHC staining for VEGF was observed in untreated SCC VII tumors. Layer-shaped staining by VEGF was seen in the remaining SCC VII tumor cells close to the necrotic area after the delivery of HT and/or the anti-angiogenic drug TNP-470. In conclusion, the injection of HT and/or TNP-470 resulted in an increase in VEGF expression. Increased VEGF expression in SCC VII tumors may be responsible for the hypoxia brought on by heat-induced vascular injury [264]. Similar results were published by Nishimura et al., where the combination of anti-angiogenic factor TNP-470 and HT (44 °C for 30 min) caused a significant tumor growth delay in comparison to one of the two therapies alone [265].

In contrast, Roca et al. reported that the anti-angiogenic cytokine plasminogen activator inhibitor-1 (PAI-1) is upregulated after HT and that its anti-angiogenic activities lessen tumor regrowth after HT treatment [266]. Interestingly, HIF-1 targets both VEGF and PAI-1. Consequently, the upregulation of HIF-1 is both pro- and anti-angiogenic [267]. The tissue response would ultimately be determined by the relative balance between the activity of these components. This does not rule out the possibility of supplementing heat with anti-angiogenic therapy to shift the balance more towards anti-angiogenesis.

## 4. Discussion

Cancer therapy is a field that is constantly evolving. Currently, IT and TT are among the most researched and sought-after compounds in the oncologic pipelines of pharmaceutical companies. While these relatively newer treatments may provide substantial benefits to patients, it remains important to simultaneously investigate synergies with already existing therapies. HT has already been known to sensitize tumor cells to CT and RT but has not been sufficiently investigated in combination with emerging therapies such as IT and TT. In this review, various effects of HT on the immune system and other important pathways have been described, which could allow researchers to identify potential modes of action that have additive or synergistic effects. To this end, three selected compounds were discussed in depth and hypotheses were generated on their potential working mechanism when combined with HT. For anti-PD-1/PD-L1 (and therefore likely ICIs in general), it appears that all HT-constituted immune mechanisms working together will play important roles in terms of efficacy. Namely, for the treatment to be effective, it is essential that tumors treated with these antibodies have a hot TME. Indeed, when the tumor is not sufficiently immune active, it has been reported that ICI therapy is not as effective since it builds on one’s own immune system. Through the influence HT has on immune cells themselves, trafficking, ICDs and other described pathways, it becomes abundantly clear that HT can turn a tumor towards this hot phenotype.

The efficacy of BiTEs seems to be dependent on several factors that HT can affect. Most importantly, increased blood flow will increase their distribution in tumors. Moreover, the addition of HT can provide a more favorable environment in terms of available TAAs/CAAs, cytokines, and activation of T-cells. Therefore, the combination of HT and BiTE molecules could also be a promising topic for further research.

The effects of the VEGF family and its receptors have been described. From the literature, it seems that HT can modulate VEGF and act in both a pro- and anti-angiogenic way, depending on the environment and possibly the thermal dose. Therefore, it could still be a viable option to explore anti-angiogenic therapies in combination with HT. In this way, the balance might be shifted towards the anti-angiogenic properties while still providing all the immunogenic and other benefits HT can bring.

## 5. Conclusions

There are several benefits that plead for the combination of HT with the above-discussed compounds. The authors acknowledge that the number of chosen compounds is currently limited and therefore will likely omit several potentially interesting targets. However, the objective of this review is to make the basis for the general mechanism of action between these types of compounds and HT. This should indeed encourage more comprehensive investigations into newly developed compounds and their interactions with critical pathways where hyperthermia also plays a role. Hopefully, such studies hold the potential to enhance the outcomes of many cancer patients and can aid in the development of more personalized medicine, considering specific genetic and molecular features of tumors and tailoring appropriate treatment strategies. Indeed, taking into account characteristics of the tumor may help in deciding on an appropriate IT or TT and dictate the fine tuning of HT parameters such as the duration, modality, and temperature. This should also encourage a shift in the existing technologies, as providing a highly accurate and precise HT treatment can be of utmost importance. Moreover, investigating the base and synergisms behind the modes of action of HT could pave the way for usage beyond the scope of cancer therapies and create an expansion towards other fields.

## Figures and Tables

**Figure 1 cancers-16-00505-f001:**
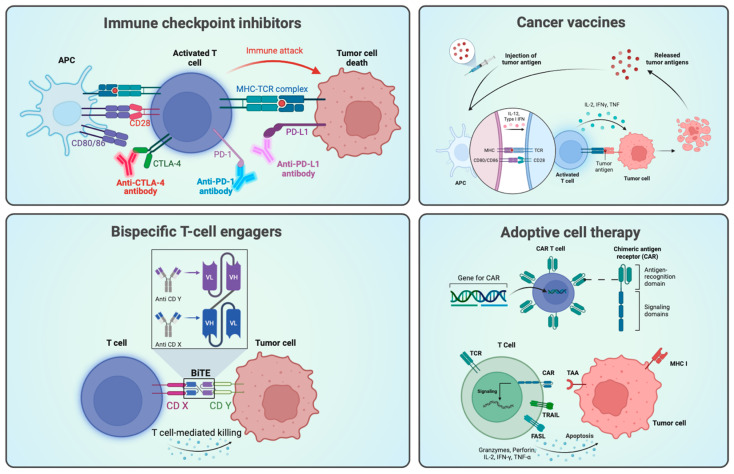
Different types of immunotherapies; figure adapted from Dagher et al. [3]. Created with BioRender.com.

**Figure 2 cancers-16-00505-f002:**
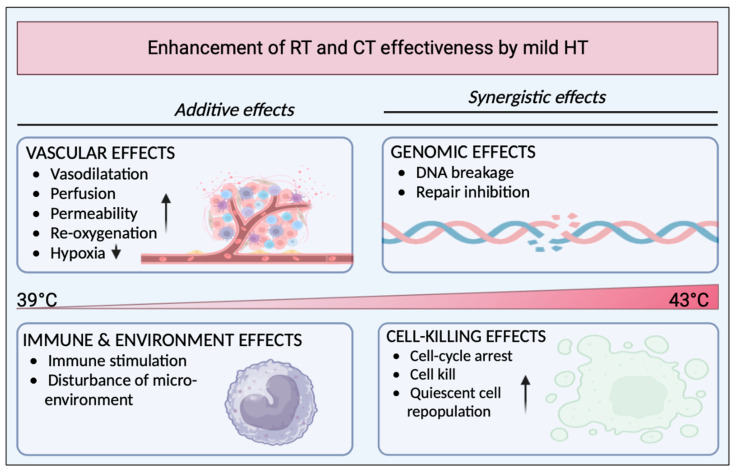
The various effects of hyperthermia enabling it to act as a radio- and chemosensitizer. Figure adapted from Oei et al. [7]. Created with BioRender.com.

**Figure 3 cancers-16-00505-f003:**
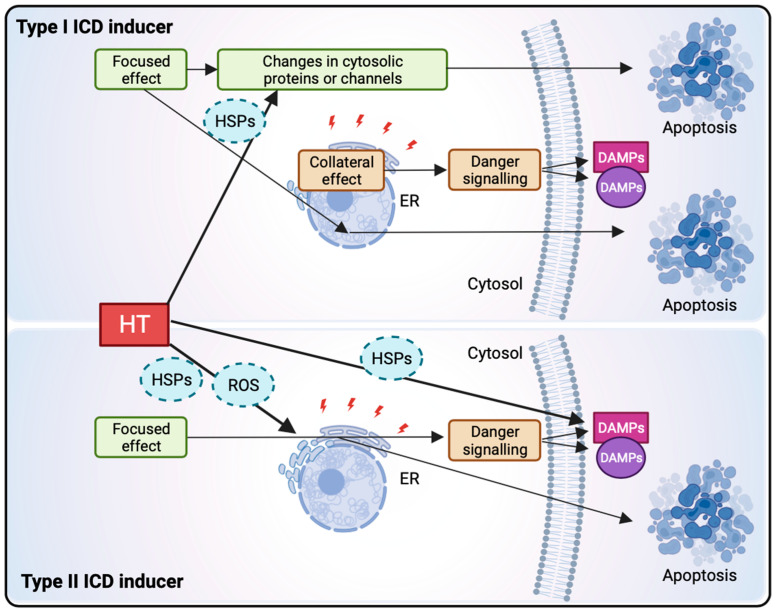
Schematic representation of two mechanisms by which ICD is induced and the influence of HT. Adapted from Krysko et al. [22]. Created with BioRender.com.

**Figure 4 cancers-16-00505-f004:**
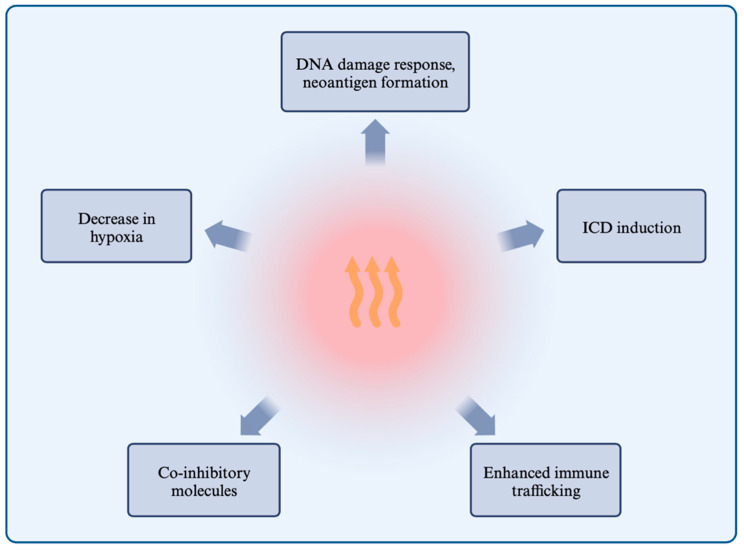
Schematic overview of HT effects on the immune system, contributing to a hot tumor environment. Created with BioRender.com.

**Figure 5 cancers-16-00505-f005:**
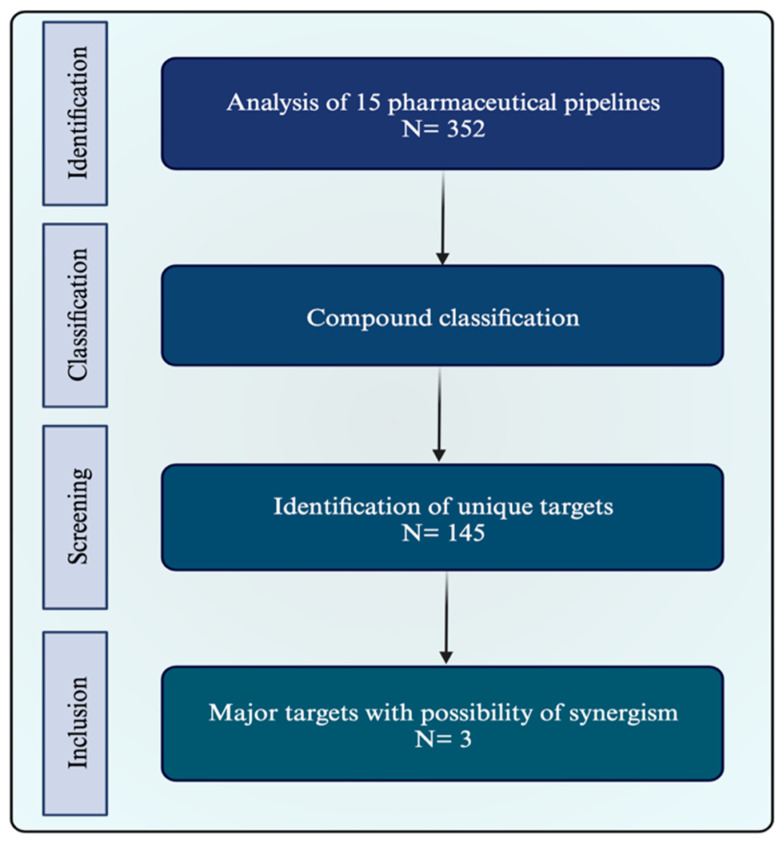
Flow chart with a schematic overview of the research strategy. Created with BioRender.com.

**Figure 6 cancers-16-00505-f006:**
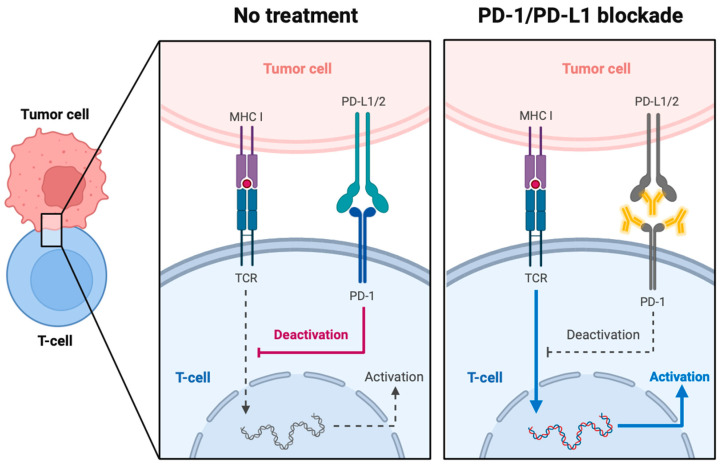
Working mechanisms of PD-1 and PD-L1 blockade in cancer therapy. Without treatment, PD-L1 expressed on tumor cells will be able to bind with PD-1 and activate the co-inhibitory signal to prevent the immune response. Using antibodies targeted against the receptor or its ligand, this interaction can be blocked in favor of immune activation. Adapted from “PD-1 Blocking Antibodies: Potential Repurposed Drug Candidate for COVID-19” by BioRender.com (2024). Retrieved from https://app.biorender.com/biorender-templates (accessed on 4 January 2024).

**Figure 7 cancers-16-00505-f007:**
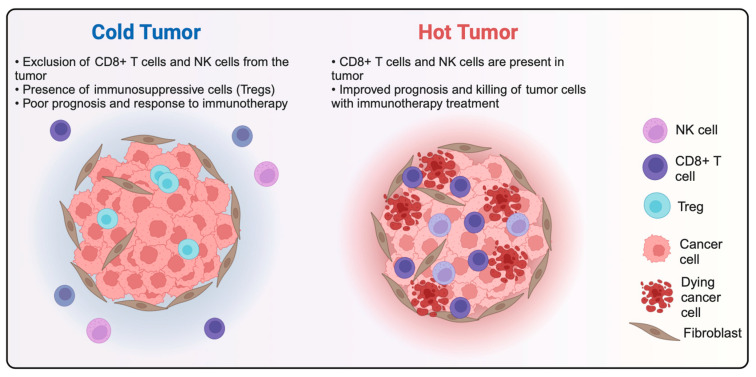
Differences between an immunologic hot tumor and cold tumor. The hot tumor displays more inflammatory cells present in the tumor, while more regulatory T-cells are seen in the cold one. The differences between the two types can account for distinct immunotherapy responses. Adapted from “Cold vs. Hot Tumors” by BioRender.com (2023). Retrieved from https://app.biorender.com/biorender-templates (accessed on 4 January 2024).

**Figure 8 cancers-16-00505-f008:**
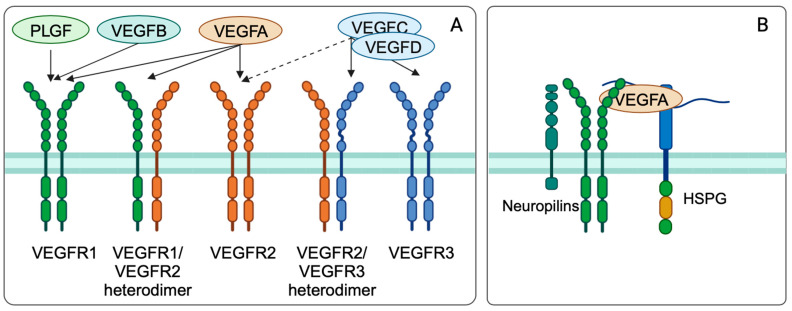
(**A**) Vascular endothelial growth factor (VEGF) ligands and receptors. The growth factors can interact with various VEGF receptors (VEGFRs), which belong to the tyrosine kinase superfamily. VEGFR1 can interact with placental growth factor (PIGF), VEGFA, and VEGFB, while VEGFR2 can interact with VEGFA, VEGFC, and VEGFD. VEGFR3 can only bind to VEGFC and VEGFD. (**B**) VEGFR co-receptors neuropilins and heparan sulfate proteoglycans (HSPGs). Figure created with BioRender.com.

**Table 1 cancers-16-00505-t001:** Overview of company pipelines.

Company	Number of Compounds Related to Solid Tumors	Trial Stage per Compound			
		I	II	III	IV
AbbVie (North Chicago, IL, USA)	16	13	1	2	
Amgen (Thousand Oaks, CA, USA)	19	13	3	3	
AstraZeneca (Cambridge, UK)	34	23	10	9	2
Bayer (Leverkusen, Germany)	10	8	2		
Bristol Myers Squibb (BMS) (New Brunswick, NJ, USA)	41	16	9	16	
GlaxoSmithKline (GSK) (London, UK)	17	8	4	5	
Ipsen (Paris, France)	3	1		2	
Johnson & Johnson (New Brunswick, NJ, USA)	12	6		4	2
Merck (Branchburg, NJ, USA)	13		13		
Merck Sharp & Dohme (MSD) (Rahway, NJ, USA)	27		16	8	3
Novartis (Basel, Switzerland)	28	13	6	9	
Pfizer (New York City, NY, USA)	31	18	5	8	
Roche (Basel, Switzerland)	75	32	5	26	12
Sanofi (Paris, France)	18	5	7	3	2
Servier (Suresnes, France)	8	6		2	

**Table 2 cancers-16-00505-t002:** FDA and EMA approvals for VEGF-related therapies.

Drug	FDA * Approval Year	EMA * Approval Year	Cancer Type
Axitinib	2012	2012	Renal cell carcinoma
Bevacizumab (Avastin)	2004	2005	Colorectal cancer, lung cancer, breast cancer, renal cancers, brain cancers, ovarian cancer, cervical cancer
Cabozantinib	2012	2016	Medullary and differentiated thyroid carcinoma, renal cell carcinoma, hepatocellular carcinoma
Everolimus	2009	2009	Advanced kidney cancer, progressive or metastatic pancreatic neuroendocrine tumors, breast cancer
Lenalidomide	2005	2008	Multiple myeloma, mantle cell lymphoma
Lenvatinib mesylate	2015	2015	Thyroid cancer, renal cell carcinoma, hepatocellular carcinoma
Pazopanib	2009	2009	Clear cell renal carcinoma
Ramucirumab	2014	2014	Gastric cancer, metastatic non-small lung carcinoma, metastatic colorectal cancer, hepatocellular carcinoma
Regorafenib	2012	2013	Metastatic colorectal cancer, advanced gastrointestinal stromal tumors, advanced hepatocellular carcinoma
Sorafenib	2005	2006	Kidney cancer, liver cancer, thyroid cancer
Sunitinib	2006	2007	Renal cell carcinoma, pancreatic neuroendocrine tumors, gastrointestinal stromal tumors
Thalidomide	2006	2009	Multiple myeloma
Vandetanib	2011	2012	Medullary thyroid cancer
Ziv-aflibercept	2012	2013	Metastatic colorectal cancer

* FDA = Food and Drug Administration. EMA = European Medicines Agency.

## Data Availability

All data used in this review from our pipeline research are made publicly available through the Appendix A.

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
