# Peer review of "Hyperthermia in Combination with Emerging Targeted and Immunotherapies as a New Approach in Cancer Treatment"

_cancers, 2024, doi:10.3390/cancers16030505_

Round 1
Reviewer 1 Report
Comments and Suggestions for Authors
Hyperthermia has long been used to cure cancer and other diseases due to its diverse origin of heating sources and noninvasiveness during treatment. It is also featured as an adjuvant therapy for other treatment modalities such as chemotherapy and radiotherapy. In addition, suitably controlled temperatures also boost the immune response and thus exhibit the potential to sensitize immunotherapy which is at the forefront of cancer therapy. In this work, Logghe et al timely summarized the combinative use of hyperthermia with targeted therapy and immunotherapy, both of which are currently in intensive investigations to cure cancer patients at advanced stages. I believe that this work can be a very valuable reference for colleagues working in diverse areas but with a focus on cancer drug development and cancer therapies, as should be published in Cancers. The manuscript is also well-organized and well-written.
Minor comments: Where appropriate, the authors should focus on the introduction of hyperthermia, such as the underlying principles, heating sources, equipment setups, detection method, and consequences, etc. The 1st part of both the Introduction and Results put on a heavy description of treatment less related to hyperthermia.
Author Response
Dear reviewer,
We would like to thank you for your time and feedback on our manuscript. Your constructive comments have helped us in refining the quality and clarity of our work. They have been addressed as follows:
- In response to the general consensus of the reviewers, we have carefully revised the manuscript, adhering to your advice to reduce the emphasis on basic immunology concepts. We have condensed the relevant sections to maintain a more streamlined and focused narrative. We believe that these changes have significantly enhanced the overall readability and accessibility of our paper. We attach the original manuscript wherein we have highlighted the sections that were removed for the revised one.
- We would like to address the concern regarding the absence of an expanded discussion on different hyperthermia equipment and techniques. While we recognize the importance of this topic in the broader context of our research, we have decided not to delve extensively into this. We believe that an in-depth exploration of hyperthermia techniques, while undoubtedly valuable, would extend beyond the intended scope of this paper. Instead, we have chosen to concentrate our efforts on providing a comprehensive and detailed analysis of synergisms between hyperthermia and emerging therapies.
Reviewer 2 Report
Comments and Suggestions for Authors
The authors of the manuscript “Hyperthermia in combination with emerging targeted- and immunotherapies as a new approach in cancer treatment” describe the role of hyperthermia as a modality as a form cancer treatment, particularly in combination with immunotherapy and targeted therapy.
The manuscript is well structured, and the topic has been extensively covered.
Figures and tables contribute to a better understanding of the signaling pathways cited in the text. Overall the paper is well crafted.
Minor criticisms:
- The font for the text used in Figure 7 and Figure 8 is small and hard to read
- In the paragraph 3.3, when vascular endothelial growth factor (VEGF) is introduced, it is rather unclear the link with hyperthermia, and a few references on the topic would help further clarifying the issue.
- There is no mention in the text nor references to the role of Insulin-like growth factor binding protein-6 (IGFBP-6) in hyperthermia and immunity.
- In the concluding paragraph (n. 5), it would be appropriate to further speculate on future developments in the field.
Author Response
Dear reviewer,
We would like to thank you for your time and feedback on our manuscript. Your constructive comments have helped us in refining the quality and clarity of our work. They have been addressed as follows:
- In response to the general consensus of the reviewers, we have carefully revised the manuscript, adhering to the advice to reduce the emphasis on basic immunology concepts. We have condensed the relevant sections to maintain a more streamlined and focused narrative. We believe that these changes have significantly enhanced the overall readability and accessibility of our paper. We attach the original manuscript wherein we have highlighted the sections that were removed for the revised one.
-
We would like to thank you for bringing the importance of IGFBP-6 to our attention, we have incorporated a mention of this protein into a relevant part of the text to emphasize this.
-
Regarding the comment on the unclear link between VEGF and hyperthermia in paragraph 3.3, we would like to draw your attention to paragraph 3.3.6, where we have addressed the link between anti-angiogenic agents and hyperthermia.
-
We have expanded the conclusion to include speculative considerations on future developments in the field. The additional text has been highlighted in the revised version.
- To improve the visual appeal and accessibility of our work, we have adapted some of the figures to enhance readability.
Reviewer 3 Report
Comments and Suggestions for Authors
Overwhelmingly comprehensive review - The message, relevance of the technology and potential benefit would come across better if the paper was more focussed. There is no need to explain basic immunology, monoclonal antibody biology etc.
Author Response
Dear reviewer,
We would like to thank you for your time and feedback on our manuscript. Your constructive comments have helped us in refining the quality and clarity of our work. We have addressed them as follows:
- In response to your general consensus, we have carefully revised the manuscript, adhering to your advice to reduce the emphasis on basic immunology concepts. We have condensed the relevant sections to maintain a more streamlined and focused narrative. We believe that these changes have significantly enhanced the overall readability and accessibility of our paper. We attach the original manuscript wherein we have highlighted the sections that were removed for the revised one.
Reviewer 4 Report
Comments and Suggestions for Authors
In the work by Tine et al., the authors present a thorough examination of the integration of hyperthermia into contemporary immunotherapies. The review is skillfully crafted, featuring clear organization and a robust rationale for proposing strategies to incorporate hyperthermia into cancer treatments. Minor change in Fig6: CD8+ T cells are the major population in targeting cancer cells which recognize MHC-I presented peptide, while CD4+ T cells recognize MHC-II.
Comments on the Quality of English Languagena
Author Response
Dear reviewer,
We would like to thank you for your time and feedback on our manuscript. Your constructive comments have helped us in refining the quality and clarity of our work. We have addressed them as follows:
- In response to the general consensus, we have carefully revised the manuscript, adhering to your advice to reduce the emphasis on basic immunology concepts. We have condensed the relevant sections to maintain a more streamlined and focused narrative. We believe that these changes have significantly enhanced the overall readability and accessibility of our paper. We attach the original manuscript wherein we have highlighted the sections that were removed for the revised one.
- The mentioned figure has been adapted.
Round 2
Reviewer 3 Report
Comments and Suggestions for Authors
comments and concerns sufficiently addressed